Critical re-examination of known purported fossil Bombycoidea (Lepidoptera)

http://orcid.org/0000-0002-9048-4381 Heikkilä Maria 1 maria.heikkila@helsinki.fi
Minet Joël 2
Zwick Andreas 3
http://orcid.org/0000-0001-5594-4154 Hundsdoerfer Anna 4
http://orcid.org/0000-0003-0937-2815 Rougerie Rodolphe 2
Kitching Ian J. 5
1 Finnish Museum of Natural History, Luomus, University of Helsinki , Helsinki , Finland
2 Institut de Systématique, Évolution, Biodiversité (ISYEB), Muséum National d’Histoire Naturelle, CNRS, EPHE, Sorbonne Université, Université des Antilles , Paris , France
3 Australian National Insect Collection, CSIRO , Canberra , Australia
4 Molecular Laboratory, Museum of Zoology, Senckenberg Natural History Collections Dresden , Dresden , Germany
5 Natural History Museum , London , United Kingdom
Gillespie Joseph
Electronic publication date: 2023 Nov 10
Publication date: 2023
Volume: 11
Electronic Location ID: e16049
Received 2023 Apr 14; Accepted 2023 Aug 16
Copyright: © 2023 Heikkilä et al.
Copyright year: 2023
Copyright holder: Heikkilä et al.
License: This is an open access article distributed under the terms of the Creative Commons Attribution License, which permits unrestricted use, distribution, reproduction and adaptation in any medium and for any purpose provided that it is properly attributed. For attribution, the original author(s), title, publication source (PeerJ) and either DOI or URL of the article must be cited.
License URL: https://creativecommons.org/licenses/by/4.0/

Keywords: Attacus? fossilis, Bombycites buechii, Bombycites oeningensis, Mioclanis shanwangiana, Sphingidites weidneri, Compression fossils, Cocoons, Cuticular fragments, Pupation chambers, Sphingid proboscis

Funding: The authors received no funding for this work.

==============================
We critically re-examine 17 records of fossils currently assigned to the lepidopteran superfamily Bombycoidea, which includes the silk moths, emperor moths and hawk moths. These records include subfossils, compression and impression fossils, permineralizations and ichnofossils. We assess whether observable morphological features warrant their confident assignment to the superfamily. None of the examined fossils displays characters that allow unequivocal identification as Sphingidae, but three fossils and a subfossil (Mioclanis shanwangiana Zhang, Sun and Zhang, 1994, two fossil larvae, and a proboscis in asphaltum) have combinations of diagnostic features that support placement in the family. The identification of a fossil pupa as Bunaeini (Saturniidae) is well supported. The other fossils that we evaluate lack definitive bombycoid and, in several cases, even lepidopteran characters. Some of these dubious fossils have been used as calibration points in earlier studies casting doubt on the resulting age estimates. All fossil specimens reliably assigned to Bombycoidea are relatively young, the earliest fossil evidence of the superfamily dating to the middle Miocene.

Introduction

The superfamily Bombycoidea is mostly diversified in the intertropical region of the globe (Kitching et al., 2018) and includes the renowned moth families Sphingidae, Saturniidae and Bombycidae. Sphingids are large pollinators with excellent flying abilities, yet important prey for bats. The tobacco hornworm Manduca sexta (Linnaeus, 1763) is a common pest sphingid species causing considerable damage to tobacco, tomato, pepper, eggplant, and plantations of other crops. Saturniids include some of the largest moth species, most famous is the giant silk moth Attacus atlas (Linnaeus, 1758) with a wingspan of 25–30 cm. The domesticated silk moth Bombyx mori Linnaeus, 1758 is a bombycid of great economic importance for silk production. Because these species have been extensively studied, they play a leading role in the fields of Lepidoptera genetics and physiology. Recently, a checklist reporting 6,092 species was provided by Kitching et al. (2018).

The Bombycoidea monophyly is corroborated by morphological and molecular data (Minet, 1994; Lemaire & Minet, 1998; Regier et al., 2008; Zwick et al., 2011; Hamilton et al., 2019). Based on molecular phylogenetics, changes were made to the higher-level classification in rapid succession. Regier et al. (2008) included Anthelidae in Bombycoidea (formerly Lasiocampoidea). Zwick (2008) synonymised the former family Lemoniidae with Brahmaeidae and re-established the bombycid subfamily Apatelodinae as a distinct family. Then Zwick et al. (2011) established Mirinidae and the former bombycid subfamilies Oberthueriinae and Prismostictinae as synonyms of Endromidae, and the former bombycine subfamily Phiditiinae as another distinct family. This resulted in the current classification that recognizes 10 families in Bombycoidea (Zwick et al., 2011; Kitching et al., 2018; Hamilton et al., 2019): Anthelidae, Apatelodidae, Bombycidae, Brahmaeidae, Carthaeidae, Endromidae, Eupterotidae, Phiditiidae, Saturniidae and Sphingidae.

Wahlberg, Wheat & Peña (2013) estimated a crown group age of 84 Ma for Bombycoidea, and Kawahara et al. (2019) one of 80 Ma. However, the fossil record of Bombycoidea is considerably younger than these estimates. The ages of the oldest fossils proposed to represent bombycoids are 53 Ma for the specimen illustrated in Grande (2013), 47.8–41.2 Ma for fossilized Saturniidae cocoons reported by Kuntz (2010), and 33.9 ± 0.1 Ma for Attacus? fossilis Cockerell, 1914 (Sohn et al., 2012). In the present work we provide arguments against the assertion that some of these fossils represent lepidopterans (see below). The oldest trace fossils attributed to Sphingidae are from the early Eocene (Roselli, 1939; Genise, Farina & Verde, 2013).

In the catalogue of fossil and subfossil Lepidoptera by Sohn et al. (2012) and Sohn, Labandeira & Davis (2015), the number of known fossil specimens placed in the superfamily Bombycoidea is estimated to be 53. However, over 37 of these are permineralized cocoons from the same site in France and initially attributed to Saturniidae, but later proposed to be pupation chambers of Hymenoptera (Kuntz, 2015). A purported saturniid fossil specimen not included in the catalogue by Sohn et al. is a compression fossil from the Green River Formation figured in Grande (2013). Other fossils not included in Sohn et al. (2012) include trace fossils (pupation chambers) found at several sites in Uruguay and Argentina and attributed to Sphingidae (Genise, Farina & Verde, 2013; Genise, 2017).

Some of the fossils listed under Bombycoidea in Sohn et al. (2012) have been used as calibration points in divergence time analyses (e.g., Kawahara & Barber, 2015). However, in many groups of Lepidoptera the original identifications of fossil specimens are known to be based on superficial similarity to modern species, not on apomorphies or reliable character combinations diagnostic of the group in question. Therefore, trusting the original identifications can lead to erroneous estimations on the age and historical biogeography of different groups of Lepidoptera. The amount of new information on the morphology and systematics of Bombycoidea, and Lepidoptera in general, has grown since the original description of many of the known fossils, thus allowing critical review of their identification.

The study at hand is part on an international collaborative project with the aim of reviewing all known fossil Lepidoptera. Reviews on the following groups have already been published: Nepticulidae (Doorenweerd et al., 2015); Papilionoidea (De Jong, 2017); Tortricidae (Heikkilä et al., 2018); Pyraloidea (Heikkilä, Simonsen & Solis, 2018), Hepialoidea (Simonsen, Wagner & Heikkilä, 2019). The objective of the present article is to re-examine known fossil Bombycoidea and discuss the information provided by reliably identified fossils of bombycoids towards our understanding of the evolutionary history and biogeography of this group.

Materials and Methods

Specimens examined

The fossils are deposited in different institutions around the world and visiting all the collections was not feasible. We were able to examine only two specimens in person: the compression fossil tentatively identified as a saturniid by Grande (2013) and examined by MH at the USNM, and the fossilized pupa identified as a bunaeine saturniid and examined by IJK when on loan to the NHMUK. Many institutions do not allow sending specimens on loan. However, we were able to obtain newly taken high-resolution photographs of several of the specimens to help us in our assessments. In these cases, the curators of the collections and the photographers were instructed as to the views and details we wished to see in close-up. We acknowledge that in such cases, and in cases when the original specimen was not located and only information in the original articles and figures was available to us, assessments could become more accurate when the original specimens are found and/or can be examined first-hand. Even so, we consider that we have been able to provide evidence and arguments for or against the placement of these fossils in Bombycoidea.

In three cases the original publication did not include a detailed description and illustrations of the specimen, and the depository was not stated. Therefore, we are unable to comment on the veracity of the identifications. These fossils are listed in Results under the subheading “Fossils not examined”.

The age estimates of the fossils were taken from Sohn et al. (2012) unless stated otherwise.

Specimen examination and character observation

The identifications of the specimens were re-evaluated by scrutiny of the visible morphological structures and assessing whether or not these provide compelling support. Explicit apomorphies that would help identify a fossil as bombycoid with more certainty are few (Lemaire & Minet, 1998: 321), and there are known exceptions to all these characters. They include:

1. Forecoxae distinctly fused anteriorly in last stage larvae (Figs. 25, 26 in Minet, 1991; not so however in Apatelodidae, Carthaeidae, most Anthelidae and certain Eupterotidae);

2. D1 setae on larval segment A8 arising from a middorsal scolus (sometimes absent or replaced by a conical protuberance; convergent evolution in some non-bombycoid families, e.g., genus Entometa Walker, 1855 in Lasiocampidae, several Notodontidae);

3. In the forewing venation, stem Rs1 + 2 closely parallel to stem Rs3 + 4 or fused to it (except in most Anthelidae);

4. Loss of the spinarea (dense group of microtrichia), which is present, ventrally, at the base of the forewing in many Lasiocampidae and indisputably belongs to the lepidopteran ground plan (although also lost, through parallel evolution, in various groups of Lepidoptera).

5. A long mesothoracic parepisternal sulcus that reaches, or terminates near, the anapleural cleft; this bombycoid autapomorphy is proposed here, based on information in Brock (1971: Figs. 38b–38d) and Minet (1994: 76). This sulcus had been regarded by Minet as a long “lower sector” of the precoxal suture (“lps”) because of Brock’s interpretation of the ditrysian mesopleurosternum (see Kristensen, 2003 (Fig. 4.17) for a correct interpretation of this region).

If we compare, in the forewing, the common stem of Rs1 and Rs2 with that of Rs3 and Rs4, the Rs1/Rs2 “forking point” is seen to lie distad of the Rs3/Rs4 forking point in many Lasiocampoidea and Bombycoidea, but this trait cannot at present be regarded as a synapomorphy of these superfamilies as it may be absent from the lasiocampid ground plan (Zolotuhin, 2010: Fig. 1, a Chionopsychinae) and from some bombycoid families (e.g., Apatelodidae). According to Hasenfuss (1999: 156), a possible synapomorphy of these two superfamilies could be the presence, in the larval proleg, of two layers of “pad cuticle” in the mesal region of the subcorona but this character remains to be verified more extensively in the Bombycoidea, having been studied in only five bombycoid families. Unfortunately, another supposed bombycoid autapomorphy in the male genitalia musculature (e.g., Minet, 1994: 71) was based on several misinterpretations in a paper by Kuznetzov & Stekolnikov (1985) and was thus rejected some years ago (Zwick, 2009).

Observing these characters in fossils is unlikely because of their often-fragmentary nature. In addition, some of the characters of interest are extremely small or are rarely, if ever, preserved because they are soft, unsclerotized structures. Because of these issues, we have also evaluated whether combinations of homoplastic characters that are typically found in Bombycoidea could be observed and tried to identify diagnostic characters of subgroups of Bombycoidea, such as families or subfamilies.

Results

The fossils are discussed under three subheadings: Fossils assigned to Bombycoidea with reasonable certainty; Fossils possibly erroneously assigned to the Bombycoidea; and Fossils not examined. When these sections include several fossils, they are discussed from oldest to youngest.

Fossils assigned to Bombycoidea with reasonable certainty:

Sphingidae

The main distinctive traits of the Sphingidae were listed by Lemaire & Minet (1998: 344). Given that Brahmaeidae and Sphingidae are no longer regarded as sister groups, we propose to add the following trait to the list of apomorphies that characterize the Sphingidae: in the hindwing venation, Sc + R is approximated to the postdiscal section of Rs (an apomorphy also present, through parallel evolution, in the Brahmaeidae).

1. Mioclanis shanwangiana Zhang, Sun & Zhang, 1994

Figure 1.

Figure 1 Mioclanis shanwangiana Zhang, Sun & Zhang, 1994.

(A) Wings as in fossil. (B) Wings drawn separately. Drawings: Joël Minet (A and B); Maria Heikkilä (C). Redrawn after Zhang, Sun & Zhang (1994). Scale bars represent: 3 mm (A and B); 5 mm (C).

Excavation data: China: Shandong, Linqu, Shanwang (Shanwang Formation); Langhian, Middle Miocene.

Depository: PFDL Shandong, China (Holotype: SK000361). We have not been able to determine where the PFDL currently is.

Published illustrations: Zhang, Sun & Zhang (1994): 82, figs. 58, 59, pl. 10: 4 (drawings).

Preservation type and size: Full-body compression/impression fossil of adult moth. A dorsal view of the fossil, in which the wings are spread slightly overlapping either side of the body, and an interpretation of the visible wing venation were illustrated in Zhang, Sun & Zhang (1994). Forewing length: ca. 22.5 mm. Fragments of proboscis, antennal bases and legs visible. Sex indeterminate.

Comments: Despite considerable effort, we were unable to obtain more information on the specimen. Assessment of this fossil is based on the illustrations and text in Zhang, Sun & Zhang (1994).

An estimated forewing length of 22.5 mm and wingspan of 45–48 mm makes Mioclanis relatively small for a sphingid but similar in size to such genera as Hemaris Dalman, 1816 and Macroglossum Scopoli, 1777.

Zhang, Sun & Zhang (1994) noted a resemblance (but also some differences) between the fossil and moths of the extant genus Clanis Hübner, 1819 (erroneously attributed to “Walker” by Zhang, Sun & Zhang, 1994), currently placed in the tribe Leucophlebiini (Sphingidae: Smerinthinae) (see Kitching et al., 2018). Thus far, the only wing trait proposed as a smerinthine apomorphy is the constriction in the forewing, some distance before the tornus, of the space between the anal vein and the inner margin (Haxaire & Minet, 2017: 111). However, this feature has been lost (= reversal) in some Smerinthinae (e.g., Leucophlebia Westwood, 1847: see Lemaire & Minet (1998: 339, fig. 18.5 I)) and so its lack in Mioclanis does not exclude this genus from Smerinthinae.

Other characters consistent with a placement of Mioclanis in Sphingidae are:

Forewing veins Rs1 and Rs2 long-stalked (or entirely fused if the very short, free Rs1 branch is an artefact). Both conditions occur in Sphingidae but the former is less common, being confirmed only in some smerinthines (e.g., Leucophlebia afra Karsch, 1891; see Lemaire & Minet, 1998: Fig. 18.5 I), Callionima parce (Fabricius, 1775) (Lima, 1950: Fig. 86), Manduca sexta (Linnaeus, 1763) (Madden, 1944: Fig. 9), Agrius cingulata (Fabricius, 1775) (Zimmerman, 1958: Fig. 377), certain specimens of Monarda oryx Druce, 1896 (Haxaire & Minet, 2017: 111) and, interestingly, Hemarini in Macroglossinae. In respect to the latter, according to the original description, the wings of Mioclanis are “translucent” (although it is not stated how this was determined), and so this character is consistent with Hemaris and Cephonodes Hübner, 1819.

Stem Rs1+2 is separate from Rs3+4 but roughly parallel to it (and very close to it). This is consistent with the usual condition in Bombycoidea, in which these stems are either closely parallel or fused together (Lemaire & Minet, 1998: 321). The only bombycoid family that does not have this feature is Anthelidae (except the antheline genus Chelepteryx Gray, 1835), in which these stems are involved in the formation of an elongate areole (= accessory cell) and so not really approximated to each other.

Forewing discal cell narrow, with its upper angle more distal than its lower angle. This is the normal sphingid condition.

Forewing vein M2 arises slightly closer to M3 than to M1 (i.e., discocellular m2-m3 = about ½ discocellular m1-m2). This again is the normal sphingid condition, although M2 arises about midway between M1 and M3 in Callionima parce (Lima, 1950: Fig. 86). However, the condition is widespread and also typical for Anthelidae and present in non-bombycoid families, e.g., some Lasiocampidae, Erebidae and Satyridae.

In both forewing and hindwing, m-cu crossvein long and in line with adjacent section of the lower edge of the discal cell. This character occurs in many Sphingidae but is relatively rare in other moth families.

Forewing anal vein distinctly arched upwards. This is typical of most Sphingidae.

Inner margin of forewing concave for much of its length. This feature is found in certain Sphingidae (e.g., Hemaris fuciformis (Linnaeus, 1758)).

Hindwing veins Rs and M1 short-stalked. This is typical of many Sphingidae but also occurs in many other moth families.

Hindwing discal cell small, elongate and roughly parallel to the costa. This distinctive shape is consistent with many Sphingidae (see, e.g., Heppner, 1998: Figs. 435 and 436).

Hindwing crossvein (R) between subcosta and upper edge of discal cell beyond half length of discal cell. In Mioclanis, hindwing crossvein (R) between Sc and the upper edge of the discal cell is more distal (beyond halfway) than in extant Sphingidae. However, although a crossing point before halfway has been claimed as a sphingid apomorphy, it does also occur in other bombycoids.

Several traits in Mioclanis disagree with the usual sphingid condition. Forewing vein Sc reaches the costa much more distally than in most sphingids, where this vein does not extend beyond the middle of the costa (e.g., Hodges, 1971). However, there are a few known exceptions, e.g., Leucophlebia afra (Lemaire & Minet, 1998: Fig. 339), Agrius cingulata (Zimmerman, 1958: Fig. 377) and Daphnis nerii (Linnaeus, 1758) (Komai et al., 2011: Fig: II-39.3 E).

In Mioclanis, forewing vein R is shown as stalked with Rs1+2. This is never found in sphingids as far as we are aware, where R arises separately from the leading edge of the discal cell around the halfway point. R is stalked with elements of the radial sector in other bombycoids. However, this may be an artefact of the drawing, given the apparent ambiguity in this region.

In Mioclanis, although Sc+R beyond the discal cell is closer to Rs than in many other moths, it is not as close to it as in most extant Sphingidae (in which vein Sc+R is distinctly approximated to the free section of Rs, at least for a short or very short distance—exceptions are rare but include the closely related genera Hemaris and Cephonodes).

The wing shape of Mioclanis is closer to some Noctuoidea.

Overall, although many characters are consistent with Mioclanis being a sphingid, none is unequivocal. Furthermore, one is completely contrary to Mioclanis being a sphingid (although consistent with some other bombycoids) and another is inconsistent with superfamily Bombycoidea. However, a comprehensive study of bombycoid wing venation is required to ensure there are no exceptions. Thus, on balance, we consider that Mioclanis probably is a sphingid but its placement within the family remains uncertain.

Mioclanis was used to provide a minimum age for the crown Smerinthini s.s. in the study by Kawahara & Barber (2015) (as 16.1 ± 0.9 Ma) and Rougerie et al. (2022).

2. Fossilized sphingid larva illustrated and described in Zeuner (1927)

Figure 2.

Figure 2 Counterpart and cast of the part of a fossilized sphingid larva (GPIT/HE/00071, NC/25/K/15).

(A) Counterpart. (B) Cast. The part has not been located at GPIT. All scale bars represent 1 cm. Photo credit: Hossein Rajaei, Staatliches Museum für Naturkunde, Stuttgart. Black and white photographs of the part and counterpart in Zeuner (1927).

Excavation data: Germany: Baden–Württemberg, Münsingen, Böttingen b. Münsingen (“Böttinger Marble”); Sarmatian, Late Middle Miocene. Excavation locality and age of deposit taken from Zeuner (1927) and specimen label, but these differ from the information given by Sohn et al. (2012).

Depository: GPIT. GPIT/HE/00071, NC/25/K/15. The counterpart and a silicone cast of the larva are in the GPIT collection. The part of this specimen has not been located (I. Werneburg, 2019, personal communication).

Published illustrations: Zeuner (1927): 321, figs. 1–3, 5 (black and white photographs). https://link.springer.com/content/pdf/10.1007%2FBF03160426.pdf

Preservation type and size: Silica or permineralization. Length: ca. 7 cm; greatest width: 1.4 cm. The larva has not been compressed and has left a concave cavity lined by a 1–2 mm thick layer of “dough-like limestone” embedded in red limestone. The head is missing, but Zeuner described the specimen as otherwise nearly complete and unusually well preserved, and with the anterior part bent upwards. The cavities left by the thoracic legs are filled with aragonite and so details cannot be observed. Details of abdominal and anal prolegs are also concealed. Zeuner noted a cavity left by a slender anal scolus (“horn”) and the anal plate is said to be relatively large with a steep orientation.

Comments: According to Zeuner, the surface ornamentation and pleats (= “annulets”) are identical to those of extant sphingid larvae. He recognized two types of sphingid larvae: (1) those in which the head capsule is rounded, the anterior three segments narrow abruptly, and the anal plate is relatively small; and (2) those in which the head is dorsally pointed, the body segments gradually narrow anteriorly, and the anal plate is large. Although the head of the fossil larva is missing, Zeuner assigned the fossil to the latter group based on the gradually narrowing body shape and a large, steep anal plate. Although annulets occur in several other lepidopteran families (Peterson, 1956), they are more numerous, 6–8 per segment, in Sphingidae, and this condition is observed here. Furthermore, the presence of only a single median scolus on abdominal segment 8 is also typical of Sphingidae, although there are exceptions (Scoble, 1992; Lemaire & Minet, 1998). However, taken together, these two features, as well as its large size, argue strongly for a placement of this fossil larva in Sphingidae, but incertae sedis because an assignment to a subfamily is too speculative.

3. Proboscis of sphingid moth in Churcher (1966)

Figure 3.

Figure 3 Proboscis of sphingid moth (right-hand lateral view). ROMIP30729. Talara Tar Pits, Talara, Peru.

© Royal Ontario Museum, Jean-Bernard Caron. Scale bar represents 1 mm.

Excavation data: Peru: Piura, Talara (Lobitos Tablazo Formation); Late Pleistocene.

Depository: ROMUT. ROMIP30729

Published illustrations: Churcher (1966): 990, fig. 15 (black and white photograph).

Preservation type and size: Coiled structure interpreted as the haustellum (proboscis) of a sphingid moth in black, asphalt-impregnated sandy matrix. The length of the structure is difficult to assess because it is coiled, and some of the coils are hidden behind others. The diameter of the coiled part of the structure (i.e., disregarding the basal (3 mm long) section) is ca. 4.2 mm. The width of the coil at the base is ca. 0.8 mm. The haustellum seems to be at least 10 cm long (by comparison with Recent Sphingidae having a coiled proboscis of a similar diameter).

Comments: The large diameter of this structure suggests it is indeed a coiled sphingid proboscis. When coiled, the well-developed proboscides of several large Erebidae (Noctuoidea) have a diameter of at most 3.5 mm (e.g., Eudocima fullonia (Clerck, 1764) and Hypopyra megalesia Mabille, 1880). The estimated length of this fossil proboscis—10–11 cm—suggests a position within the Sphinginae, the only sphingid subfamily in which proboscides of this length have been recorded (Miller, 1997; Ryckewaert et al., 2011).

4. Fossil larva reported by Leakey (1952) and identified as a possible sphingid by Kitching & Sadler (2011)

Figure 4.

Figure 4 Cast of fossil larva (KNMI-MW 261) reported by Leakey (1952) and identified as a possible sphingid by Kitching & Sadler (2011).

Fossil specimen not located. Photo credit: Job Kibii, NMK.

Excavation data: Kenya: South Nyanza, Rusinga and M’fwangano Islands in Lake Victoria (Hiwegi Formation); Burdigalian, Early Miocene.

Depository: British-Kenya Miocene Expedition Collection, NMK. Accession No. KNMI-MW 261. The specimen was not located but a cast of it was found.

Published illustrations: Leakey (1952): 624, fig. 1 (black and white photograph).

Preservation type and size: Silica or permineralization. Whole body of a larva. The fossil has retained the three-dimensional shape of the larva. Length 4 cm, width 0.7 cm.

Comments: Kitching & Sadler (2011) wrote “Leakey (1952) illustrated an apparently large lepidopteran larva from the early Miocene deposits on Rusinga and Mfangano Islands in Lake Victoria, Kenya. The general smooth shape and secondary annulations of the body suggest this fossil may belong to the family Sphingidae (hawkmoths), although it lacks the anal horn typical of larvae of that family”.

The actual specimen was not located but we were able examine the cast of the fossil by means of 3D photogrammetry and colour photographic images provided by Job Kibii, Stephen Maikweki and Francis Muchemi (NMK), but have been unable to reach any more definite conclusions. A broken-off anal horn is unlikely in life (although they are sometimes bitten off in captivity when larvae are overcrowded and some species do lack them in the final instar), but it is possible the horn was broken off from the fossil, especially if the preparator was not expecting it. The short prolegs suggest it is a “macrolepidopteran” but the head appears large, relative to the prothorax rather than the body diameter, and the anal segment seems somewhat modified and deflected downward, features that suggest it could be Hesperiidae (D. Wagner, 2019, personal communication). Furthermore, the anal prolegs are relatively small, which is not the condition normally found in Sphingidae, and the annulets, though present, are neither obvious nor numerous. Overall, therefore, while it remains possible that this fossil is a sphingid, other “macrolepidopteran” families cannot be ruled out and the family identification must be considered incertae sedis.

Saturniidae

Although Minet (1994: 83) proposed seven apomorphies for the characterization of the Saturniidae (e.g., tarsomere 4 of the foreleg sexually dimorphic, with a pair of distal, tooth-like structures in the female), it should be noted that all of them belong to the imaginal stage.

5. Fossilized pupa discussed and illustrated by Kitching & Sadler (2011)

Figure 5.

Figure 5 Fossilized pupa from Laetoli, Tanzania. (EP 352/03).

Late Pliocene. (A) Ventral view. Arrows pointing at antenna and labial palps. (B) Lateral view. (C) Dorsal view. (D) Oblique dorsal view of abdominal segment 10 showing the shallow L-shaped groove (arrow). (E) Posterior view showing radial supporting struts (arrow) around posterior margin of abdominal segment 7. (F) Close-up of mesonotal and metanotal calli. Scale bars represent: 5 mm (A–C). Photo credit: The Trustees of the Natural History Museum, London, UK.

Excavation data: Tanzania: Laetoli, Upper Laetoli Beds (Laetoli Formation); ?Gelasian, Late Pliocene.

Depository: NMT. EP 352/03.

Published illustrations: Kitching & Sadler (2011): 551–552, figs. 20.1a–c, g–h (black and white photographs).

Preservation type and size: Permineralization. Pupa, whole body male. Length 37 mm; width 15 mm; depth 11 mm. The authors describe the fossil as slightly compressed dorsoventrally. A detailed description was given by Kitching & Sadler (2011).

Comments: Kitching & Sadler (2011) identified this fossil as a pupa of a saturniid moth in the tribe Bunaeini (Bunaeinae Bouvier, 1927 according to Nässig, Naumann & Oberprieler (2015) and Rougerie et al. (2022)), a tribe exclusively Afrotropical in distribution. The authors compared the fossil with several extant species of Bunaeini. The closest resemblance was found to be with the pupa of Cirina forda (Westwood, 1849), although the fossil was not identified as this but a species near it. The authors also acknowledged that the reference material available at the NHMUK (twelve species from nine genera) was far from comprehensive and with many species not examined, there could be other species that fit equally well or better.

The characters that Kitching & Sadler (2011) stated as supporting placement of the fossil in Bunaeini include radial supporting struts around posterior margins of abdominal segments 2 and 3 dorsally and around the entire circumference of segment 7, and a pair of shallow L-shaped grooves on the dorsum of abdominal segment 10. The “radial supporting abdominal struts” match character 17 proposed as an autapomorphy of the tribe by Rougerie & Estradel (2008): junction zone between A2/A3, A3/A4, and A7/A8–10 highly sclerotized with a row of numerous vertical grooves. Dorsal grooves (or more developed cavities) were found to be present in all the Bunaeini examined by Rougerie & Estradel (2008, their character 18), but also in most Micragonini and Urotini. In addition, the fossil pupa has the characteristic elevated crest on the posterior margin of A4–A6 (character 16 of Rougerie & Estradel, 2008) found in the vast majority of Bunaeini and which is only observed outside Bunaeini in the genus Usta Wallengren, 1863 of tribe Urotini. It gives a unique aspect to the fossil pupa (as seen in fig. 20.1b of Kitching & Sadler, 2011), in which it appears more obvious than on the live pupa of Cirina forda illustrated in Kitching & Sadler (2011).

In their article on Bunaeini, Rougerie & Estradel (2008) separated a group of four genera (Pseudobunaea Bouvier, 1927; Athletes Karsch, 1896; Lobobunaea Packard, 1901 and Pseudimbrasia Rougeot, 1962) based on the configuration of appendages on the cephalic mask of the pupa, and in particular the antennae being far from reaching the midline of the pupa. In contrast, in all other examined Bunaeini, including Cirina, the antennae reach the midline, with only the maxillae or small parts of thoracic legs visible. In Fig. 6, it is clear that the antennae of the fossil are short and the appendages are clearly visible (maxillae, legs), whereas in the illustrated Cirina pupa in Kitching & Sadler (2011), the antennae clearly meet medially. These characters indicate that the fossil is not Cirina, and also exclude several other genera within the tribe.

Figure 6 Attacus? fossilis Cockerell, 1914 (as cf. Rothschildia fossilis in Sohn et al., 2012). UCM-8554.

Photo credit: David Zelagin, UCM. Scale bar represents 5 mm.

Thus, while the identification of the fossil as Bunaeini is well supported, the genus-level identification needs further study.

In their divergence time study, Kawahara & Barber (2015) used this fossil to determine the minimum age of Cirina forda as 3.66 Ma.

Fossils possibly erroneously assigned to the Bombycoidea:

6. Trace fossils of alleged sphingid or saturniid pupation chambers in the ichnogenus Teisseirei Roselli, 1939

Excavation data: Specimens interpreted as representing the ichnotaxon Teisseirei have been found in the Early Eocene Asencio Formation, Uruguay (see Genise, Farina & Verde, 2013); localities of different Cenozoic ages in Argentina (Puerto Unzué Formation, Gran Salitral Formation, Sarmiento Formation, see Genise, Farina & Verde (2013) and references therein, and the middle Miocene Collón Curá Formation at El Petiso, Chubut province, see Genise et al., 2022); and the Pliocene deposits at Laetoli, Tanzania (see Genise & Harrison, 2018).

Depository: The material examined by Genise, Farina & Verde (2013) is deposited in the following collections: Colección de Icnología del Museo Argentino de Ciencias Naturales, Buenos Aires (MACN-Icn); Museo Paleontológico Egidio Feruglio Trelew, Chubut, Argentina (MPEF-Ic); and Colección Paleontológica de la Facultad de Ciencias, Montevideo, Uruguay (FCDPI). Material examined by Genise & Harrison (2018) is deposited in the Harrison collection; and the material examined by Genise et al. (2022) is in Ichnological Collection of the Museo Paleontológico “Egidio Feruglio”, Trelew, Chubut province, Argentina (MPEF-IC).

Published illustrations: Teisseirei barattinia Roselli, 1939: Roselli (1939): 82, figs 29 and 30 (drawings); 84, fig. 31:7 (black and white photograph); Melchor, Genise & Miquel (2002): 25, figs. 12 A–E, I (black and white photographs); Genise (2004): 431, figs. 3 b, c (black and white photographs); Genise, Farina & Verde (2013): 481, fig. 1 (colour photographs) https://doi.org/10.1111/let.12025; Genise (2017): 346, figs. 13.25; 349, figs. 13.28 a–d (colour photographs). Teisseirei linguatus Genise & Harrison, 2018: Genise & Harrison (2018): 604, fig. 5 C–J (colour photographs); Teisseirei barattinia and Teisseirei paladinco Genise & Cantil, 2022: Genise et al. (2022): 10–11, figs. 7 A–I and 8 A (colour photographs).

Preservation type and size: Trace fossils. There is some variation among the numerous specimens of the Teisseirei ichnospecies, but in general they constitute of horizontal to sub-horizontal chambers (enlargements of burrows) with a depressed, elliptical cross-section, antechamber and multi-layered lining and inner surface covered in densely spaced sub-rectangular or sub-triangular pits. On some of the chambers, a thin, discrete wall can be observed. Internal casts of the chambers have also been found. For an amended diagnosis of the ichnogenus Teisseirei, see Genise et al. (2022).

The size ranges of the several hundred chambers examined by Genise, Farina & Verde (2013), Genise et al. (2022) and Genise & Harrison (2018) were as follows: length—1.9–9.1 cm; width—0.9–4.9 cm; and height—0.75–3 cm. One exceptionally large chamber was 11.5 cm long and 7 cm wide. Genise, Farina & Verde (2013) suggested that the variation could be mostly taphonomic, but because the structures are from different localities, it is also possible, even likely that different species produced them.

Comments: Originally, these structures (“Teisseirei barattinia”) were suggested to be pupation chambers of Hymenoptera (Roselli, 1939). Later, they were tentatively associated with Coleoptera (Roselli, 1987; Genise, 2004). A new hypothesis that they were sphingid pupation chambers was proposed by Genise, Farina & Verde (2013), who made macro- and micromorphological comparisons of these structures to pupation chambers burrowed by larvae of the modern sphingid species Manduca rustica (Fabricius, 1775) and Eumorpha labruscae (Linnaeus, 1758), and observed similarities. In particular, the authors emphasized the similarity in the distinct type of multi-layered lining of the chambers, which they interpreted to be the result of the larva packing soil dampened by liquid it had excreted. The densely pitted internal surface texture visible in Teisseirei barattinia specimens was also found to be similar to that seen inside M. rustica pupation chambers. The pits were interpreted to be imprints of thoracic legs. The authors also hypothesized that the antechamber of T. barattinia and the hatch in modern pupation chambers through which the adult emerges, could be comparable in function. Because pupation in M. rustica and E. labruscae does not occur very deep in the soil, the trace fossils were suggested to serve as indicators of uppermost horizons of palaeosols (Genise, Farina & Verde, 2013). However, Genise, Farina & Verde (2013) did note that in addition to Sphingidae, subterranean pupation chambers are also known in other Lepidoptera, such as Noctuidae, Geometridae, and Saturniidae, but the features and differences among these have not been thoroughly studied.

After the description of other ichnospecies in the ichnogenus Teisseirei, Genise et al. (2022) amended the diagnosis of the ichnogenus and now attributed Teisseirei ichnospecies to the pupation chambers of both Sphingidae and Saturniidae. Ichnotaxa are based on the fossilized work of organisms but although the nomenclature of ichnotaxa resembles the conventional Linnean system of classification, an ichnotaxon can include specimens that resemble each other in morphology but those characteristics are not necessarily to be interpreted as evidence of a shared most-recent common ancestor. The ichnogenus Teisseirei belongs in the ichnofamily Coprinisphaeridae; other ichnogenera in that ichnofamily are attributed to Coleoptera, Hemiptera and Hymenoptera (Genise, 2004; Genise et al., 2022).

We consider that a ca. 2 cm long chamber, the minimum size mentioned by Genise, Farina & Verde (2013), is too small for a sphingid or a saturniid pupation chamber. According to Bell & Scott (1937: 341), the smallest known hawkmoth pupa (that of the Tiny Hawkmoth, Sphingonaepiopsis pumilio (Boisduval, 1875)) is 20 mm long. They add that it lies in a “rough cocoon” that is not subterranean—and this cocoon must necessarily be longer than 20 mm. Furthermore, to the best of our knowledge, no recent Sphingidae or Saturniidae pupation chambers have “antechambers”. Thus, we consider it impossible at present to be certain that these pupation chambers were made by sphingid or saturniid larvae specifically, rather than by the larvae of other lepidopteran families (and possibly even other insect orders). There are hundreds of specimens placed in the ichnogenus Teisseirei. It is possible that some of these fossil chambers are trace fossils produced by Sphingidae or Saturniidae, but it is also entirely possible that most of them may eventually prove not to be lepidopteran at all.

7. Fossilized ovoid structures reported by Kuntz (2010)

Excavation data: France: Alsace, North Middle Upper Rhine Graben, Bouxwiller quarry (Bouxwiller Formation); Lutetian, Middle Eocene.

Depository: The depository was not given in Kuntz (2015) but in Kuntz (2010) he implies that such fossils are in several museum and private collections. Sohn et al. (2012) stated that the specimens are deposited in “various institutes”, but these were not listed. The exact number of specimens is not given.

Published illustrations: Kuntz (2010): Figs. 40–45 (photographs); Kuntz (2015) (colour photographs) https://asam67.org/bouxwiller-2015-les-ovoides-ont-de-nouveaux-parents/.

Preservation type and size: Permineralized ovoid structures proposed to be fossilized cocoons. The length of the largest of these ovoid specimens ranges from 5.5 to 7 cm, and the diameter from 2.5 to 3 cm. One extremity of these structures is rounded, the other pointed or flared. The surface is uneven, with imprints likened to crossing silk fibers. Some specimens have a slight dent in the middle of the long side along with a stronger calcification, possibly attesting a horizontal position of the cocoon with respect to the ground. Many of these cocoons have an opening, which Kuntz interpreted as the hole from which the adult moth had emerged.

Comments: Sohn et al. (2012) listed these specimens in fossil Saturniidae following Kuntz (2010), who proposed that they were the cocoons of saturniid moths. The main evidence he gave to support this view were the flared openings at one extremity of some of these structures, which he interpreted as similar to the cocoons of Saturniidae such as Saturnia pavonia (Linnaeus, 1758) in which the narrower, somewhat open anterior end has an internal ring of apically convergent stiffer “bristles” that serve to prevent ingress of predators while facilitating the emergence of the adult moth. In addition, the surface of the fossils seems to have an irregular, slightly helical, striped pattern that is perpendicular to the long axis of the cocoon. Kuntz considered this type of texture to be somewhat similar to that on cocoons spun by many recent saturniids, with embossing on the surface formed by crossing silk fibers. However, in his 2015 publication, Kuntz concluded that these egg-shaped structures are more likely pupal chambers of spider wasps, such as those of the genus Pepsis Fabricius, 1804 (Pompilidae) (guêpe géante) (Kuntz, 2015). The size, the apparent solidity and the more or less helical striation was proposed to support this hypothesis, but the variable shape of the opening was problematic. Kuntz supposed the shape of the opening could help in the attribution of these egg-shaped structures to an insect group, but he also noted that the shape could be related to the stage of eclosion at the moment of fossilization.

We agree that these are most probably not fossilized lepidopteran cocoons.

8. Attacus? fossilis Cockerell, 1914 (as cf. Rothschildia fossilis in Sohn et al., 2012)

Figure 6.

Excavation data: USA: Colorado, Teller County, Florissant Beds National Monument, Florissant Formation; Late Priabonian, Late Eocene (33.9 ± 0.1 Ma).

Depository: UCM. Holotype: UCM-8554.

Published illustrations: Cockerell (1914): 271, fig. 34 (drawing).

Preservation type and size: A compression fossil with what Cockerell (1914) interpreted as the imprint of the apex of the forewing with veins of a large moth in the family Saturniidae (Fig. 6). The fragment is 33 mm in length.

Comments: The fossil shows at least five more or less parallel arched lines, some of which are incomplete. The distance between the arched lines is about 5 mm. There are no obvious stalked or connate veins, and no traces of a wing pattern or scales. Cockerell (1914) interpreted the parallel arched lines as veins, and the shorter line in the lower right of the fragment (as viewed in fig. 8), more or less perpendicular to the longest vein, as a short segment of the wing margin (see fig. 34 in Cockerell, 1914). Cockerell considered the venation of the fossil to closely correspond to that of the forewing of Attacus dohertyi Rothschild, 1895, and tentatively named the specimen Attacus? fossilis. In the catalogue by Sohn et al. (2012), the specimen is referred to as cf. Rothschildia fossilis following Schüssler (1933), who transferred “fossilis” from Attacus to the genus Rothschildia Grote, 1896, probably because the former does not occur in the New World. Below we attempt to reconstruct the reasons and characters that presumably led Cockerell to assign the fossil to Saturniidae. We also evaluate whether these characters can reliably place the fossil in this family.

The longest of the veins on the fossil was interpreted by Cockerell as vein “R5”, (i.e., Rs4 in current venation nomenclature), and he considered that the rather strongly curved shape of the veins and the arrangement of Rs4 in relation to the short wing margin section resembled the distal (apical) part of the forewing of certain Saturniidae. The strongly arched veins Rs4 and M1 indeed occur in the tribe Attacini but also in some Antheraea Hübner, 1819 (see fig. 92 in Michener, 1952) and several Arsenurinae (see, e.g., fig. 40 (Caio richardsoni (Druce, 1890), fig. 41 (Rhescyntis pseudomartii Lemaire, 1975) in Michener (1952), and figs. 56, 57 and 126 in Lemaire (1980)). The relatively greater distance separating Rs4 from the vein below (M1) could also have been seen as a feature found in large Lepidoptera, such as saturniids. In addition, the concave shape of the wing margin at the apex of Rs4 occurs occasionally in Rhescyntis Hübner, 1819 (Lemaire, 1980: Fig. 126) but practically never in Antheraea and Rothschildia. In contrast, the oblique line of M2 (the short, incomplete vein below M1) would fit better with Saturniinae (e.g., Antheraea) than with Arsenurinae.

We compared the veins on the fossil with those of several species of extant large saturniid moths (those mentioned above and figures in Rougerie (2005)) by superimposing the fossil veins onto illustrations of their forewing venation. In many cases the curvature of the veins was too strong and did not correspond to that of the extant species. However, the curvature did follow more closely the veins of the extant species of Attacini and Antheraea, but otherwise there was no other obvious support for an assignment to the Saturniidae.

We also asked paleobotanist Dr Herbert Meyer (Florissant Fossil Beds National Monument, Colorado, USA) and paleoentomologist Dr Conrad Labandeira (NMNH, Washington, D.C., USA) to examine a photograph of the fossil. They concluded that the imprint on the slab was probably made by a leaf. This assessment was based on the observation that the line considered by Cockerell to be a short segment of the wing margin was actually the thicker primary vein of a leaf. The arched veins (Cockerell’s R and M veins) were interpreted as secondary veins of the leaf. The secondaries were also noted to merge into the primary and not end abruptly as would be expected in an insect. Possible plant genera candidates could be Staphylea L., Hydrangea L., or Celastrus L. (H. Meyer, 2016, personal communication).

Attacus? fossilis was used as a calibration point in the divergence time analysis by Kawahara & Barber (2015) to give a minimum age to the stem group of Rothschildia and Saturnia Schrank, 1802. The supporting information of their study stated that the fossil shares synapomorphies with extant Rothschildia and Saturnia, a mistake the authors were not able to correct after the final edits (A. Kawahara, 2015, personal communication). Given the very different interpretations of the fossil, we conclude that the identification is based on superficial similarity and additional characters would be needed to place it reliably in Saturniidae (or any of the proposed plant genera, for that matter).

9. Compression-impression fossil of adult moth in Zhang (1989)

Figure 7.

Figure 7 (A and B) Compression-impression fossil of adult “sphingid” moth first illustrated in Zhang (1989). no. 820157.

Photo credit: Sun Mingchang, SFML.

Excavation data: China: Shandong, Linqu, Shanwang (Shanwang Formation); Langhian, Middle Miocene.

Depository: SFML. no. 820157.

Published illustrations: Zhang (1989): 94, pl. 20: 3 (black and white photo).

Preservation type and size: Compression-impression fossil of an adult moth. Poorly preserved. Head, thorax, abdomen, left forewing and base of right forewing partly visible. Abdominal segments with impressions of hair-like scales of reddish-brown colour. Some wing venation visible on wings. Length of left forewing about 2.3 cm. Length of the preserved part of the body is 25.2 mm. Width of abdomen at its widest part 1 cm.

Comments: Zhang (1989) identified the fossil as a sphingid based mostly on forewing characteristics but noted that the genus and species cannot be determined. Zhang wrote that the fossil has some similarities to moths in the genus Clanis Hübner, 1819 (misattributed to Walker by Zhang (1989)) but did not elaborate on these. According to the original description by Zhang (1989), the forewing veins Rs3 and Rs4 (cited just as R and R) are stalked, M1 (cited as just M) originates in the upper corner of the discal cell, and Sc, R, Rs1 and Rs2 (cited as Sc, R1, R2 and R3) are parallel and closely aligned. Five abdominal segments can be distinguished. However, only part of the forewing venation is visible in the specimen and the above description by Zhang (1989) is inaccurate. Importantly, vein M2 is straight and arises midway between M1 and M3, a character that suggests this fossil differs from Mioclanis shanwangiana and may even not belong to the Sphingidae (in which vein M2 arises closer to M3 than to M2; Lemaire & Minet (1998)). The poor preservation of the specimen and lack of characters does not allow a reliable identification of this specimen to superfamily level (or lower).

10. Sphingidites weidneri Kernbach, 1967

Excavation data: Germany: Lower Saxony, Willershausen am Harz; Piacenzian, Late Pliocene.

Depository: GZG. Holotype: GZG.W.03445 (old no. 596-11). The specimen has not been located but is most certainly in the GZG collection (A. Gehler, 2018, personal communication). We were able to examine a photograph of the original photograph by Adolf Straus, used by Kernbach and published in the “Berichte der Naturhistorischen Gesellschaft Hannover” (1967). There is a typographical error in A. Straus’s specimen number in Kernbach (1967) where it was given as 3435. In the photograph presented in Kernbach’s publication, the specimen number had been cropped so that it cannot be completely seen. The complete number is 3445.

Published illustrations: Kernbach (1967): 108, fig. 11 (black and white photograph) https://www.zobodat.at/pdf/Ber-Nathist-Ges-Hannover_111_0103-0108.pdf.

Preservation type and size: Whole body compression-impression fossil of a larva. Size not given by Kernbach (1967).

Comments: Brauckmann, Brauckmann & Gröning (2001) considered Kernbach’s description of the genus Sphingidites to be invalid because of the lack of a diagnosis. However, Sohn & Lamas (2013) supported the interpretation that Kernbach intended this genus to accommodate fossil Sphingidae whose association below family-level is not convincing and thus, as a collective genus, no type species or diagnosis is required. A subsequent type designation had been provided by Clark et al. (1971: 582) but this was also unnecessary because the type would have been automatically fixed by monotypy. The circumscription of the genus is not affected by the type species designated by Clark et al. (1971).

Kernbach (1967) interpreted the specimen to be probably a (prepupal) larva whose transformation from larva to pupa had been disturbed. He reported the presence of several larval segments and an anal horn. Some transverse lines are visible in the photograph that could be interpreted as larval segments and a darker, narrow and short projection at one end of the fossil, the possible anal horn, can be observed. However, because these characters are not very clear and others cannot be made out, we agree with Kozlov (1988: 23, 55) and consider the identification of this fossil as a sphingid to be uncertain. Indeed, it is very difficult to interpret and possibly does not even represent a caterpillar.

11. Bombycites oeningensis Heer, 1849

Figure 8.

Figure 8 Bombycites oeningensis Heer, 1849.

One of the abdomens is three lines (6.3 mm) wide and six lines (12.6 mm) long, the other 2.5 lines (5.25 mm) wide and 5.5 lines (11.5 mm) long (1 line = 2.1 mm). Photograph of illustration in original publication. The publication is no longer under copyright.

Excavation data: Germany, Baden-Württemberg: Oeningen (“Molasseformation”), that is Wangen (near Öhningen—see e.g., Cockerell, 1915); Messinian, Late Miocene.

Depository: Heer (1849) wrote that the specimen is deposited at the University of Zurich and according to Sohn et al. (2012) the holotype is in the PIMUZ. However, it is not in the PIMUZ database (https://www.pim.uzh.ch/apps/cms/pageframes/sammlung_db.php), which includes all published specimens (C. Klug, PIMUZ, 2018, personal communication). It was not found in the ETH Zürich, Earth Science Collections (or database) either, where most holotypes described by Heer are deposited (A. Mueller, 2018, personal communication).

Published illustrations: The article was first published as a separate in 1849 (Heer, 1849) but also again the following year in Heer (1850). The same illustration (drawing) was included in both publications: Heer, 1849: 183, pl. XIV: fig. 7; and Heer, 1850: pl. XIV, fig. 7. See Biodiversity Heritage Library: https://biodiversitylibrary.org/page/2477621.

Preservation type and size: A compression-impression fossil of two very fragmentary adult moths. According to Heer (1849), the abdomens and fragments of the wings are visible. One of the abdomens is three lines (6.3 mm) wide and six lines (12.6 mm) long, the other 2.5 lines (5.25 mm) wide and 5.5 lines (11.5 mm) long (1 line = 2.1 mm). Heer speculated that the wider abdomen belonged to a female moth, the narrower to a male of the same species. No details of the wing venation or wing shape can be made out.

Comments: Heer (1849) referred to these fossils as “Noctuo-Bombycida” and did not even narrow the identification down further to “Bombyces”.

Both Handlirsch (1906–1908) and Kozlov (1988) placed the specimen in the category of Lepidoptera incertae sedis. We agree with that assessment as no characters presented in the illustration or described in the original publication enable placing of the moths in any lepidopteran superfamily. Even the identification of the depicted impressions as moths is difficult. Handlirsch (1906–1908) stated “pupa” as the stage of the fossil, which is understandable because it is not obvious that the illustration provided by Heer (1849, 1850) represents two adult moths.

The name “Bombycites” was first used by Latreille (1817: 561) for a suprageneric group (“tribe”) within recent “Phalaenae” (i.e., moths). It was proposed as a generic name—Bombycites—by Heer (1849: 183), of which the type-species is the quite enigmatic Bombycites oeningensis Heer, 1849 (Fletcher & Nye, 1982). It was later used for a collective group aimed at accommodating fossils proposed to be bombycoids but for which a genus-level identification is not possible (Heer, 1865; Sohn & Lamas, 2013).

12. Bombycites buechii Heer, 1865

Figure 9.

Figure 9 Bombycites buechii Heer, 1865. Specimen barcode number: 0000000005466.

Scale bar represents 2 mm. Photo credit: Earth Science Collections of ETH Zürich.

Excavation data: Germany, Baden-Württemberg: Oeningen (“Molasseformation”) (i.e., Wangen); Messinian, Late Miocene.

Depository: ETH. Specimen barcode number: 0000000005466.

Published illustrations: Heer (1865): 397, fig. 310 (drawing).

Preservation type and size: Compression-Impression fossil of a larva (whole body). Length of larva ~4 cm, width at widest part ~1.3 mm. The larva seems to be in lateral view.

Comments: The lack of details in the original description and diagnostic characters led Kozlov (1988) to place the specimen in his list of Papilionida (i.e., Lepidoptera) incertae sedis. We agree that the identification of this fossil as a bombycoid is very uncertain. It is possibly not even a larva (there seems to be an elongate, tapering appendage (antenna ?) adjacent to it, but admittedly not necessarily part of this fossil). In addition, there are no obvious prolegs. This is perhaps not even an insect.

13. Compression-impression fossil of wing scale tentatively assigned to a sphingid moth by George (1952)

Excavation data: Pakistan: Punjab, Salt Range, Warcha and Jankush Nulla Gorges (Saline Series dolomite); Late Eocene.

Depository: SJCA Uttar Pradesh; slide no. 16. We have been unable to reach the curator in charge of the collection to request a new photograph of the specimen.

Published illustrations: George (1952): 88, fig. 55 (drawing). We have been unable to reach the editors of this journal to request permission to reproduce the original image.

Preservation type and size: Compression/impression fossil of a wing scale of an adult moth. The drawing shows a long and narrow scale, bent and folded close to its mid-length. The scale has longitudinal striations, and the apex has three shallow subtriangular lobes. The total length is described to be 640 micra (μm) and the width at the widest part about 64 micra (μm).

Comments: The author stated that “the unmistakable sphingid facies can be made out” but no additional details to support this assessment were provided. No comprehensive study of lepidopteran wing scales has yet been done and we are unaware of characters that would unambiguously and definitively assign a wing scale to Sphingidae. We agree with Kozlov (1988), who placed this specimen in the category of uncertain identifications.

14. Fossilized scales and cuticular fragments in gut contents of fossil bats in Richter & Storch (1980)

Excavation data: Germany: Hesse, S Frankfurt, near Darmstadt, Messel oil shale-layers (Messel Formation); Early Lutetian, Middle Eocene.

Depository: SF.

Published illustrations: Richter & Storch (1980): 365, fig. 16, but see Comments below. We have been unable to reach the editors of this journal to request permission to reproduce the original image.

Preservation type and size: Fossilized scales and cuticular fragments of Lepidoptera in the gut contents of fossilized bats. SEM images presented in Richter & Storch (1980) reveal that the microstructure of the scales has been preserved well. Cuticular fragments are small and do not contain diagnostic structures such as legs, antennae or larger hollow structures that have been compressed. Association of the cuticular fragments with body parts is difficult, except for wing fragments (double-layer of cuticle). These cuticular wing fragments show detailed sculpturing, including a more or less dense cover of trichomes (“false hairs”) in the case of lepidopteran wings.

Comments: Sohn et al. (2012) stated that Fig. 16 in Richter & Storch (1980: 365) could be a possible sphingid scale, probably because it is very similar to the scales of modern Sphingidae figured by Richter & Storch (1980: Fig. 17). However, Richter & Storch (1980) said that this type of scale, i.e., with inter-ridge perforations and cross-ridges, is typical of many lepidopteran families, including Sphingidae, Noctuidae and Saturniidae. Assigning such lepidopteran scales to a particular family is indeed difficult because such microstructure can be observed in many groups of the Coelolepida (Lepidoptera with hollow scales) (Kristensen & Simonsen, 2003; van Eldijk et al., 2018). In addition, the shape and structure of lepidopteran scales can vary even on the same wing, and they are thus not very informative phylogenetically (Kristensen & Simonsen, 2003). Some of the scales in the gut contents are said to show similarities to those of modern Cossidae, Micropterigidae and Eriocraniidae, the latter two of which are mostly diurnal, unlike bats. The abundance of cuticular fragments with trichomes led Richter & Storch (1980: 365) to the conclusion that the dominant prey of these bats had been small, “primitive” Lepidoptera, because wings with trichomes between scales are known from the families Micropterigidae, Eriocraniidae and Hepialidae. There is no evidence that would indicate the cuticular fragments or scales to belong to Sphingidae or any other bombycoid family. On the contrary, based on the absence of certain scale types, Richter & Storch (1980: 364) even concluded that Lasiocampidae were not part of the gut contents.

15. Non-lepidopteran fossil insect erroneously assigned to Saturniidae by Grande (2013)

Figure 10.

Figure 10 “Dayvault specimen”. USNM PAL 618360.

(A) Compression fossil erroneously identified as a saturniid in Grande (2013). (B) Detail showing numerous crossveins. Scalebar represents 1 cm (A). Photo Credit: Alan Rulis, USNM and Maria Heikkilä.

Excavation data: USA: Wyoming, Lincoln County, Green River Formation, Fossil Butte Member, locality F; Ypresian, Eocene. According to Grande (2013), the fossil lake sediments were deposited about 53–51 Ma.

Depository: Originally, the fossil was part of the private collection of the late Richard D. Dayvault but was donated to the USNM in 2016 by his wife, Jalena Dayvault. USNM PAL 618360, part and counterpart labeled A and B.

Published illustrations: Grande (2013): 76, fig. 33 (colour photograph).

Preservation type and size: Compression fossil of a winged insect in lateral aspect. Forewing length ~5 cm.

Comments: A closer inspection of the venation of this insect immediately reveals that it is not a lepidopteran. There are more veins (crossveins, notably) than in the wings of either Trichoptera or Lepidoptera (Fig. 10B, close-up showing the crossveins). The venation is reticulate and appears more similar to that of, e.g., Orthoptera or Neuroptera. We are currently unaware if any progress regarding the identification of this fossil has been made. Mrs Jalena Dayvault, who donated the specimen to the USNM, has expressed the wish that, if possible, the scientific name to be given to this specimen should somehow incorporate ‘Dayvault’, in memory of her husband. We will leave the description of this specimen to those with more knowledge of the group of insects that it represents.

16. Fossils of non-lepidopteran insects and a crustacean erroneously assigned to Sphinx:

Myrmicium schroeteri (Germar, 1839) (Sphinx schroeteri Germar, 1839 and Sphinx snelleni Weyenbergh, 1869) and the Sphinx larva illustrated by Weyenbergh (1869)

Figure 11.

Figure 11 Fossils of non-lepidopteran insects and a crustacean erroneously assigned to Sphinx.

(A) Sphinx schroeteri Germar, 1839. MB.I.860. Photo downloaded from https://portal.museumfuernaturkunde.berlin/ License: CC0. (B) Sphinx larva described in Weyenbergh (1869). 15403. Photo credit: Teylers Museum, Haarlem, Netherlands. (C and D) Sphinx snelleni Weyenbergh, 1869. 15396 and 15397. Photo credit: Teylers Museum, Haarlem, Netherlands. All scale bars represent 1 cm.

Excavation data: Germany: Solnhofen limestone deposits in Bavaria (Altmühltal Formation); Tithonian (150.8–145.5 Ma), Upper Jurassic.

Depository: Sphinx snelleni (Weyenbergh, 1869): TMH. 15396 and 15397; and “Sphinx larva” 15403. Myrmicium schroeteri (Germar, 1939): MfN. MB.I.0860.

Published illustrations: Sphinx schroeteri Germar, 1839: Schröter (1784) Plate III, fig. 16 https://zs.thulb.uni-jena.de/rsc/viewer/jportal_derivate_00164692/NLKN_1784_Bd01_%200593.tif?logicalDiv=jportal_jparticle_00152562 (drawing); https://portal.museumfuernaturkunde.berlin/detail/0d66f2851d77db8ebdf9 (colour photograph).

Sphinx snelleni Weyenbergh, 1869: Weyenbergh (1869): Plate I, fig. 9. https://www.biodiversitylibrary.org/page/24004107 (drawing);

Sphinx larva: Weyenbergh (1869): Plate I, fig. 10. https://www.biodiversitylibrary.org/page/24004107 (drawing) and Wikimedia Commons https://commons.wikimedia.org/wiki/File:Myrmicium_snelleni_Teylers_museum.jpg (colour photograph).

Preservation type: Compression fossils.

Comments: Sphinx snelleni was described by Weyenbergh (1869). The fossil is illustrated in Plate I, fig. 9 of this publication along with another fossil labelled as a Sphinx larva (Plate I, fig. 10). The original description of Sphinx snelleni mentions a coiled proboscis (which is also clearly shown in the corresponding figure: Pl. 1, fig. 9), a trait that suggests that this taxon could indeed belong to the Lepidoptera (perhaps even the Sphingidae). A curved structure is indeed also visible in photographs of the specimen, but it is difficult to interpret whether it really is a proboscis. After examination of the larval specimen, Handlirsch (1906–1908) concluded that it was the abdomen of a decapod (Crustacea). Sphinx snelleni was identified as a wood wasp of the hymenopteran family Siricidae. However, it was later moved to Pseudosiricidae as a junior synonym of what is now Myrmicium schroeteri (originally described as “Sphinx schröteri” by Germar (1839)). For more references, see Sohn et al. (2012).

17. Fossilized flower petal of Nymphaea tentatively interpreted as a sphingid larva by Nel & Nel (1985)

Excavation data: France: Les Figons, Aix-en-Provence; Rupelian, Oligocene.

Depository: MNHN. n°215 A

Published illustrations: Nel & Nel (1985) 126, figs. 11, 12.

Preservation type and size: Compression fossil. Length 2 cm.

Comments: Subsequently, the specimen and additional material were carefully reexamined by Dr. André Nel. He concluded that they are fossilized water lily petals (Sohn et al., 2012; A. Nel, 2023, personal communication).

Fossils not examined:

Sphingid in Baltic amber mentioned by Berendt (1830)

Excavation data: Baltic Region (Baltic Amber, Prussian Fm.); Lutetian, Middle Eocene.

Depository: An important part of the Berendt amber collection is in the MfN, but the specimen Berendt identified as “Sphinx” has not been located. There is no specimen in the MfN labelled as such (T. Léger, 2019, personal communication).

Published illustrations: none.

Preservation type and size: Specimen in Baltic amber. Berendt does not specify if the inclusion in amber is an adult or a caterpillar. However, the way the text is written implies it is a caterpillar. Condition and size unknown.

Comments: Berendt (1830: 36–37) mentioned a “Sphinx” in Baltic Amber. From the text it cannot unambiguously be determined whether the specimen was an adult or caterpillar: “Lepidopteren finden sich am seltensten. Ich besitze nur einen Sphinx von bedeutender Grösse. Kleine Raupen sieht man öfter” (Translation: Lepidoptera are the rarest. I only own a single Sphinx of significant size. Small caterpillars can be seen more often). The way the statement is phrased implies that it is a caterpillar of significant size whereas the others he has seen are small.

Taken at face value, this fossil would represent the oldest evidence of Bombycoidea. However, the identification cannot be confirmed because the specimen has not been located and is not described in sufficient detail in the original publication. Kusnezov (1941: 69) possibly had access to this specimen and identified the inclusion as a lepidopteran but did not suggest a lower-level identification.

Compression-impression fossil of a sphingid larva and a poorly preserved “Bombyx” mentioned by Schöberlin (1888)

Excavation data: Switzerland: Neuchâtel Canton, Oeningen (“Stinkschiefe”)/Messinian, Late Miocene.

Depository: The larva was originally in the (private?) Massmann Collection (Sohn et al., 2012), but its current depository is unknown. The whereabouts of the poorly preserved “Bombyx” fossil is not known either. We were unable to examine these specimens.

Published illustrations: none.

Preservation type and size: Compression/Impression fossil of a larva (whole body) and a poorly preserved “Bombyx” fossil (2 species?). Size not given in Schöberlin (1888).

Comments: The author likened the size of the fossil larva to that of the larva of the extant species Hemaris fuciformis (Linnaeus, 1758). Because of the lack of details and illustrations in the original publication, and the unavailability of the specimens for closer examination, their assignment to Bombycoidea cannot be confirmed. In addition, back in 1888, “Bombyx” would have been used for any “Bombyces”, i.e., including Bombycoidea (except Sphingidae), Notodontidae, Erebidae (subfamilies Lymantriinae and Arctiinae), Limacodidae, Zygaenidae and Psychidae. Thus, the mention of a “Bombyx” fossil does not necessarily mean that it belongs to Bombycoidea in the current sense, it could have been just about anything (see, e.g., Packard (1893) for an example of what was then considered to belong to “Bombyces”).

Thoracic segment of Aglia tau (Agliinae) larva in sieved residue (Lindberg, 1900).

Excavation data: Finland: Lohja; Pleistocene.

Depository: not known.

Published illustrations: none.

Preservation type and size: First thoracic segment of larva. Size not known.

Comments: Lindberg (1900) gave credit for the identification of the specimen to Finnish entomologist, Enzio Reuter. According to the information in Lindberg (1900), the segment had well-preserved “strange” horn-like structures typical of Aglia tau (Linnaeus, 1758). These are probably the scoli found on the thoracic segments of early instar Aglia larvae. There are several recent species in the genus Aglia (Kitching et al., 2018) of which only Aglia tau occurs in present day Finland.

Compression-impression fossil identified as Sphinx by Haase (1890)

Excavation data: Excavation data or depository not known.

Depository: Originally in private collection of A. Assmann. According to information found online (https://en.wikipedia.org/wiki/August_Assmann, accessed 17.03.2020), the Assmann collection is nowadays in NHUW, Wrocław. However, the entomology collection at NHUW does not include compression/impression fossils, and Assmann’s specimens are probably not in the collection of the NHUW paleontology department either, which has only vertebrates (M. Wanat, 2020, personal communication).

Published illustrations: none.

Preservation type and size: Compression/Impression fossil. Size not known.

Comments: Haase (1890: 26) mentioned that he had seen a drawing of the specimen shown to him by Mr A. Assmann. According to Haase, Assmann had intentions to publish on the specimen. The location of the specimen was not given. Handlirsch (1906–1908: 628) wrote that he was not able to locate it either and that to his knowledge Assmann’s descriptions of these fossils were not published.

Discussion

The re-examination of the 17 records shows that only five fossils can be placed in Bombycoidea with reasonable certainty—4 to Sphingidae and 1 to Saturniidae (see Table S1). However, none of the four fossil sphingids displays unequivocal characters and their identification as Sphingidae is not 100% certain. This precludes their use as calibration points according to the criteria proposed by Parham et al. (2012). Furthermore, the use of some of the dubious fossils as calibration points in earlier studies (e.g., Attacus? fossilis in the study on the hawkmoth radiation by Kawahara & Barber (2015)) casts doubt on the resulting ages. New analyses with revised sets of fossils or calibration times would be welcome in these cases.

Although all known bombycoid fossils examined are relatively young, the oldest is Mioclanis shanwangiana from middle Miocene, the origin of the superfamily is expected to be significantly older. In studies focusing on all Lepidoptera, Wahlberg, Wheat & Peña (2013) and Kawahara et al. (2019) estimated a crown-group age of 84 Ma (95% HPD: 74–93) and 80 Ma (95% HPD: 70–90) for Bombycoidea, respectively. In a study on Saturniidae, Rougerie et al. (2022) estimated the stem age of the family to be in the early Cenozoic at about 63 Ma (95% HPD: 59–69 Ma). We note however that the estimate by Wahlberg, Wheat & Peña (2013) used time calibrations derived from a set of fossils that included some that have now been shown to be misidentified, while the selection of fossils in the studies by Kawahara et al. (2019) and Rougerie et al. (2022) were based on stricter criteria.

Unfortunately, bombycoid moths, as lepidopterans in general, are rare in the fossil record (Labandeira & Sepkoski, 1993; Sohn et al., 2012), and therefore, estimates of their age and evolution remain mostly based on the combination of molecular data and secondary calibrations. The probable reason for the scarcity of fossil Lepidoptera is that scales are water-repellent, thus preventing specimens from sinking to the bottom of water bodies where they would have been buried in sediment (Martínez-Delclòs, Briggs & Peñalver, 2004; Peñalver & Grimaldi, 2006). A relatively high body-fat content of bombycoids may also increase buoyancy (Simonsen, Wagner & Heikkilä, 2019). The majority of fossil Lepidoptera are amber inclusions but nearly all of these are small moths (Sohn, Labandeira & Davis, 2015). Large moths are extremely rare as amber inclusions, and a reason may be that scales are relatively easily lost and doing so prevents big moths from getting trapped in amber. Large dead moths are also an attractive food source to scavengers and so may get spotted and eaten before they can be fossilized.

Conclusions

Our study is a contribution to efforts to obtain a more reliable and accurate understanding of the evolutionary history and historical biogeography of Lepidoptera. We critically re-examined 17 records of fossils currently assigned to the lepidopteran superfamily Bombycoidea, and assessed whether observable morphological features warrant their confident assignment to the superfamily.

The study confirms that the identifications of many of the known fossil Bombycoidea were based on overall similarity to extant species and not apomorphies. None of the examined fossils displays characters that allow unequivocal identification as Sphingidae, but three fossils and a subfossil (Mioclanis shanwangiana Zhang, Sun & Zhang, 1994, two fossil larvae, and a proboscis in asphaltum) have combinations of diagnostic features that support placement in the family. The identification of a fossil pupa as Bunaeini (Saturniidae) is well supported. The other fossils that we evaluate lack definitive bombycoid and, in several cases, even lepidopteran characters.

We can only hope that new discoveries of well-preserved fossil Bombycoidea will be made in the future and can reveal more on the evolutionary history of these moths and allow corroboration or critical revision of the current estimates of their ages.

Supplemental Information

Supplemental Information 1 Information on the repository, age, and reliability of identification of the examined fossil Bombycoidea.

Click here for additional data file.

Daniil Aristov (Paleontological Institute, Russian Academy of Sciences), Chenyang Cai (Nanjing Institute of Geology and Palaeontology), Lance Grande (The Field Museum, Chicago), Jalena Dayvault, Alexander Gehler (GZG), Dale Greenwalt (Smithsonian Institution, NMNH), Talia Karim (UCM), Job Kibii (NMK), Christian Klug (PIMUZ), David Kohls, Conrad Labandeira (Smithsonian Institution, NMNH), Théo Léger (MfN), Stephen Maikweki (NMK), Finnegan Marsh (Smithsonian Institution, NMNH), Hebert Meyer and students (Florissant Fossil Beds National Monument), Sun Mingchang (SPFL), Francis Muchemi (NMK), Andreas Müller (ETH Zürich), Francis Ndiritu (NMK), André Nel (MNHN), Hossein Rajaei (State Museum of Natural History Stuttgart), Alexandr Rasnitsyn (Paleontological Institute, Russian Academy of Sciences), Michael Rasser (State Museum of Natural History Stuttgart), Anne Schulp (TMH), Jennifer Strotman (Smithsonian Institution, NMNH), Marek Wanat (NHUW), Ingmar Werneburg (GPIT), Huang Ying and Jukka Tabell, Tim de Zeeuw (TMH), David Zelagin (UCM).

Institutional abbreviations

ETH ETH Zürich, Earth Science Collections, (= ETH Zürich, Erdwissenschaftliche Sammlungen) Zurich, Switzerland

GPIT Palaeontological Collection of Tübingen University (= Geologisch-Palaeontologisches Institut Tübingen), Tübingen, Germany

GZG Geoscience Centre of the University of Göttingen, Göttingen, Germany (= Geowissenschaftliches Zentrum der Georg-August-Universität, Geowissenschaftliches Museum) Göttingen, Germany

MfN Museum für Naturkunde—Leibniz Institute for Evolution and Biodiversity, Berlin, Germany

MNHN National Museum of Natural History, Paleontology (= Muséum National d’Histoire Naturelle, Paléontologie), Paris, France

NHMUK Department of Palaeontology, Natural History Museum, London, United Kingdom

NHUW Museum of Natural History at University of Wroclaw (= Muzeum Przyrodnicze we Wrocławiu), Wroclaw, Poland

NMK National Museums of Kenya, Nairobi, Kenya

NMT National Museum of Tanzania, Dar es Salaam, Tanzania

PFDL Paleontological Fossil Depository (= 山东临朐山旺古生物化石保护管理所), Linqu, Shandong, China

PIMUZ Paleontological Institute and Museum, University of Zurich (= Paläontologisches Institut und Museum, Universität Zürich), Zurich, Switzerland

ROMUT Royal Ontario Museum, University of Toronto, Toronto, Canada

SF Senckenberg Research Institute and Natural History Museum (= Senckenberg Forschungsinstitut und Naturmuseum Frankfurt), Frankfurt, Germany

SFML Shanwang Fossil Museum (= 山东临朐山旺古生物化石博物馆), Linqu, Shandong, China

SJCA St. John’s College, Agra, Uttar Pradesh, India

TMH Teylers Museum, Haarlem, The Netherlands

UCM University of Colorado Museum of Natural History, Boulder, Colorado, U.S.A.

USNM United States National Museum of Natural History, Washington, DC, U.S.A.

Additional Information and Declarations

Competing Interests

Author Contributions

Data Availability

The authors declare that they have no competing interests.

Maria Heikkilä conceived and designed the experiments, performed the experiments, analyzed the data, prepared figures and/or tables, authored or reviewed drafts of the article, and approved the final draft.

Joël Minet conceived and designed the experiments, performed the experiments, analyzed the data, prepared figures and/or tables, authored or reviewed drafts of the article, and approved the final draft.

Andreas Zwick performed the experiments, analyzed the data, authored or reviewed drafts of the article, and approved the final draft.

Anna Hundsdoerfer performed the experiments, analyzed the data, prepared figures and/or tables, authored or reviewed drafts of the article, and approved the final draft.

Rodolphe Rougerie performed the experiments, analyzed the data, authored or reviewed drafts of the article, and approved the final draft.

Ian J. Kitching conceived and designed the experiments, performed the experiments, analyzed the data, prepared figures and/or tables, authored or reviewed drafts of the article, and approved the final draft.

The following information was supplied regarding data availability:

There is now raw data or code to submit.

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
