# Peer review of "Critical re-examination of known purported fossil Bombycoidea (Lepidoptera)"

_PeerJ, doi:10.7717/peerj.16049_

## Round 0.1 · original submission · Minor Revisions

Dear Dr. Heikkilä and colleagues:

Thanks for submitting your manuscript to PeerJ. I have now received two independent reviews of your work, and as you will see, the reviewers raised some relatively minor concerns about the research. This is great and indicates optimism for your work and the potential impact it will have on research studying Bombycoidea systematics.

There appears to be some helpful advice on how to improve the analysis and presentation of your study. Also, please note that reviewer 1 has provided a marked-up version of your manuscript.

Therefore, I am recommending that you revise your manuscript, accordingly, taking into account all of the issues raised by the reviewers. I do believe that your manuscript will be ready for publication once these issues are addressed.

Good luck with your revision,

-joe

·

Basic reporting

The manuscript is very clearly written, its objective unambiguously stated, and results given in an objective way (as possible, given the subject matter). The literature was obviously closely reviewed and the authors apparently tried quite extensively to trace original depositions of fossil specimens. The images are as clear as they can be, again considering the circumstances of preservation or difficulty of access. Overall the paper is a valuable contribution.

Experimental design

No comment.

Validity of the findings

In some cases, the underlying data (the fossils) could not be figured nor examined by the authors, but in those cases, the authors clearly stated that they could not assign systematic placement of the fossils in question. Such hesitance is vital in the case of fossil identifications so as to preclude them from use in future divergence time estimation, pending any revised identification.

Additional comments

I found a few minor corrections in the PDF which I am uploading. There seemed to be a number of cases where the URLs led to the incorrect place or to dead links. I ask the authors to double-check all links, or perhaps allocate them to the references or a supplemental file since the embedded URLs are a bit distracting.

Reviewer 2 ·

Basic reporting

no comment

Experimental design

no comment

Validity of the findings

no comment

Additional comments

This paper provides several valuable pieces of information on the fossil record of Bombycoidea. Excluding the fossils misidentified or poorly identified would be necessary for fossil-calibrated divergence estimations. I have no doubt that the research community will get benefits from this study.
There are a few gentle suggestions to maximize the benefits for future researchers who may find new fossils.
- The authors provided the characteristics of Bombycoidea but not the subordinate family groups. The authors are undoubtedly the world leading researchers for this superfamily. So, their statements about the identification of the bombycoid families will help the subsequent researchers to ID their fossil specimens.
- It is regretful that the authors have not examined the actual specimen of Mioclanis, given its importance for the fossil record of Bombycoidea. Can the authors contact any relevant Chinese researcher and get more information about the specimen? The authors stated that it is "probably" a sphingid. Then, Zeuner's (1927) record is the only fossil whose sphingid association is supported with certainty. Right?
- The order of the species accounts seems confusing. Is it according to the age of each fossil account? Can those be arranged in family level first and then in age order?
- I recommend that the authors provide a table to summarize their review of all fossils (including reliability of identification, age etc).
- In the names of wing veins such as Rs4 and M1, the numbers should be subscripts.

---

## Round 0.2 · accepted · Accept

Dear Dr. Heikkilä and colleagues:

Thanks for revising your manuscript based on the concerns raised by the reviewers. I now believe that your manuscript is suitable for publication. Congratulations! I look forward to seeing this work in print, and I anticipate it being an important resource for groups studying Bombycoidea systematics. Thanks again for choosing PeerJ to publish such important work.

Best,

-joe

Reviewer 2 ·

Basic reporting

no comment

Experimental design

no comment

Validity of the findings

no comment